# Nutritional Composition and Bioactive Properties of Wild Edible Mushrooms from Native *Nothofagus* Patagonian Forests

**DOI:** 10.3390/foods11213516

**Published:** 2022-11-04

**Authors:** Maximiliano Rugolo, Rafael Mascoloti Spréa, Maria Inês Dias, Tânia C. S. P. Pires, Mikel Añibarro-Ortega, Carolina Barroetaveña, Cristina Caleja, Lillian Barros

**Affiliations:** 1CONICET/Centro de Investigación y Extensión Forestal Andino Patagónico (CIEFAP), Ruta 259 km 3.4, Esquel 9200, Chubut, Argentina; 2Centro de Investigação de Montanha (CIMO), Instituto Politécnico de Bragança, Campus de Santa Apolónia, 5300-253 Bragança, Portugal; 3Labortório Associado para a Sustentabilidade e Tecnologia em Regiões de Montanha (SusTEC), Instituto Politécnico de Bragança, Campus de Santa Apolónia, 5300-253 Bragança, Portugal

**Keywords:** non-timber forest products, metabolites, functional food, antioxidant properties, antimicrobial activity

## Abstract

*Nothofagus* forests of the Andean Patagonian region are home to numerous wild edible mushroom (WEM) species with interesting organoleptic characteristics, although many of them have unknown nutritional and nutraceutical profiles. The proximal composition, fatty and organic acids, soluble sugars, phenolic compounds, ergosterol, as well as antioxidant and antimicrobial activity of 17 WEMs were analyzed. Carbohydrates, the most abundant macronutrients, varied between 49.00 g/100 g dw (*C. magellanicus*) and 89.70 g/100 g dw (*F. antarctica*). Significantly higher values were found for total fat in *G. gargal* (5.90 g/100 g dw) followed by *A. vitellinus* (4.70 g/100 g dw); for crude protein in *L. perlatum* (36.60 g/100 g dw) followed by *L. nuda* (30.30 g/100 g dw); and for energy in *G. gargal* (398 Kcal/100g) and *C. hariotii* (392 Kcal/100g). The most effective extracts regarding the TBARS antioxidant capacity were those of *Ramaria*. This is the first time that a study was carried out on the chemical composition of *G. sordulenta*, *C. xiphidipus*, *F. pumiliae*, and *L. perlatum*. The promotion of sustainable use of WEMs, including their incorporation in functional diets that choose WEMs as nutritious, safe, and healthy foods, and their use in an identity mycogastronomy linked to tourism development, requires the detailed and precise nutritional and nutraceutical information of each species.

## 1. Introduction

Edible wild mushrooms are highly available functional foods. Its consumption has been developed and perpetuated in various countries from all over the world [1]. Their commercial and culinary importance is mainly due to their organoleptic properties, such as aroma and flavor, their nutritional qualities, and their medicinal characteristics [2,3,4,5], due to their high protein and fiber content, essential amino acids, bioactive compounds, and low lipids content [3,6].

Different mushrooms have been studied in search of new therapeutic alternatives, finding that they have bioactive properties [7,8,9] and that they constitute rich sources of nutraceuticals molecules [10,11], which are responsible for their antioxidant [12,13,14] and antitumor properties [15]. Antioxidants from edible natural products are currently widely studied for their ability to protect organisms and cells from damage caused by oxidative stress, which is one of the causes of aging and degenerative diseases [16].

Patagonian Andean forests comprise 3,240,996 h dominated by *Nothofagus* spp. (*N. antarctica*, *N. dombeyi*, and *N. pumilio* are the most representative species) [17]. The region harbors numerous species of wild fungi that are potentially edible, with high nutritional and medicinal value [18,19]. These species have been mentioned by Mapuche communities as having continuity of use over time, excellent organoleptic properties, and commercially viable [20]. The list of prominent and understudied species with consumption records includes *Aleurodiscus vitellinus* (Lev.) Pat., *Cyclocybe aegerita* (V. Brig.) Vizzini, *Cyttaria hariotii* E. Fisch., *Cortinarius magellanicus* complex Speg., *Cortinarius xiphidipus* M.M. Moser y E. Horak, *Fistulina antarctica* Speg., *Fistulina endoxantha* Speg., *Fistulina pumiliae* González, Barroetaveña & Pildain, *Flammulina velutipes* (Curtis) Singer, *Grifola gargal* Singer, *Grifola sordulenta* (Mont.) Singer, *Hydropus dusenii* (Bres.) Singer, *Lepista nuda* (Bull.) Cooke, *Lycoperdon perlatum* Pers., *Ramaria botrytis* (Pers.) Bourdot and *R. patagonica* (Speg.) Corner, and *Pleurotus ostreatus* (Jacq.) P. Kumm. [20,21,22,23,24]. Some of these and other cultivated fungi from Argentina and Chile were recently studied to check their nutritional and antioxidant potential [14,25]. However, the list of analyzed WEMs in this regard is not complete. New techniques are available to recheck and compare antioxidant contents, along with other bioactive properties such as antimicrobial activity [26], and compositions from specimens from different areas and habitats of already analyzed species should be carried out to know their variability [27]. Wild fungi have gained special interest in recent decades due to their value as functional foods and their promising future related to the development of local economies [19,28,29]. The combined study of their taxonomy and molecular genetic diversity [19,30,31,32,33,34,35], their phenology, ecology, and productivity [19,22], and the determination of their nutritional and nutraceutical profiles are required to expand the current variety of harvested species (mainly *Morchella* spp. and *Suillus luteus* [20]) for their sustainable and safe use, thereby creating novel modalities of products and services that locals could offer [28].

This study aimed to widen the comprehension of the biological properties and nutritional composition of endemic and cosmopolitan species of wild Patagonian edible mushrooms growing in *Nothofagus* forests. The bioactivity evaluation focused on phenolic compounds contents and antimicrobial and antioxidant properties. The chemical analysis comprised the determination of macronutrients (protein, lipids, carbohydrates, and ashes) and the composition of sugars, fatty acids, and organic acids. Comparison with previous reports for some species are included.

## 2. Materials and Methods

### 2.1. Fungi Identification and Sampling

Specimens of seventeen species (Figure 1) of WEMs (*Aleurodiscus vitellinus, Cyclocybe aegerita, Cyttaria harioti, Cortinarius magellanicus, Cortinarius xiphidipus, Fistulina antarctica, F. endoxantha, F. pumiliae, Flammulina velutipes, Grifola gargal, G. sordulenta, Hydropus dusenii, Lepista nuda, Lycoperdon perlatum, Pleurotus ostreatus, Ramaria botrytis*, and *R. patagonica*) were sampled during the mushroom fruiting seasons in Patagonia: fall (April–May) and spring (October–November) of 2019 and 2020, according to species phenology [22]. Locations included *Nothofagus* spp., *Maytenus boaria*, and *Lomatia hirsuta* forests from National Parks of the Chubut (PN Los Alerces and PN Lago Puelo), Río Negro (PN Nahuel Huapi), and Neuquén (PN Lanín) provinces, Argentina. Each sample (complete fruitbodies) was freeze-dried, pulverized, and stored in polyethylene bags at −18 °C in a freezer for subsequent analyses. Representative species of each species were dehydrated and incorporated in the Herbarium of the Patagonian Forest Research Center (CIEFAP; Esquel, Chubut, Argentina).

### 2.2. Nutritional Characterization

Samples of each species were analyzed for nutritional composition (protein, carbohydrates, fat, ash, and energy) using AOAC procedures [36]. The total carbohydrates were obtained by difference, and the energy values were calculated with the equation Energy (kcal) = 4 × (g protein + g carbohydrates) + 9 × (g fat). The results are expressed in kcal per 100 g of dry weight (dw).

### 2.3. Chemical Composition

#### 2.3.1. Free Sugars

Free sugars determination followed the methodology by Barros et al. [37]. Analysis was performed by liquid chromatography (HPLC, Knauer, Smartline 1000 systems, Berlin, Germany), coupled with a refraction index detector (Knauer Smartline 2300). The detected compounds were identified by comparison with the retention times of the standards. Trehalose was used as the internal standard. Results are expressed in g/100 g of dry weight (dw).

#### 2.3.2. Fatty Acids

The fatty acids were identified by gas chromatography with flame ionization detection (GC-FID), as previously described by Pereira et al. [38]. The identification of fatty acids was made according to their relative retention times of the FAME peaks of the sample standards (mixture 37, 47885-U purchased from Sigma). To process the results, we used CSW 1.7 software (DataApex 1.7, Prague, Czech Republic); results are expressed as a relative percentage (%).

#### 2.3.3. Ergosterol

The ergosterol was quantified after extraction following Vieira Junior et al. [39]. It was determined by high-performance liquid chromatography (HPLC) coupled to a UV detector (280 nm), as described by Cardoso et al. [40], and was identified and quantified by comparison with the pure chemical standard and expressed in mg/100g dw.

#### 2.3.4. Organic Acids Composition

The organic acids were determined by high-performance liquid chromatography coupled to a photodiode detector (UFLC-PDA) following the methodology described by Barros et al. [37]. The detection of organic acids was achieved using a DAD system, applying a wavelength of 215 nm (and 245 nm for ascorbic acid). The quantification was carried out by comparing the area of their recorded peaks with the calibration curves obtained from the standards of the respective compound. The results are expressed in mg/100 g (fw).

#### 2.3.5. Phenolic Composition

Samples of 0.5 g of freeze-dried specimens were used for the extract preparation. They were initially macerated at room temperature with the addition of a solution (30 mL) of ethanol/water (80:20, *v*/*v*), for 1 h (150 rpm). Ethanol was removed under reduced pressure. Afterwards, the aqueous phase of both extracts was frozen and lyophilized.

The identification and quantification of the phenolic compounds followed the previously optimized methodology [41], using a Dionex Ultimate 3000 UPLC system (Thermo Scientific, San Jose, CA, USA). The DAD and mass spectrometer (LTQ XL mass spectrometer, Thermo Finnigan, San Jose, CA, USA) were working in negative mode.

### 2.4. Bioactivities Evaluation

#### 2.4.1. Evaluation of Antioxidant Activity

An in vitro assay based on the monitoring of malondialdehyde (MDA)-TBA complexes was carried out as previously reported [42] to measure the extract capacity to inhibit the formation of thiobarbituric acid reactive substances (TBARS). Porcine brain cells were used as biological substrates. The results are expressed as IC_50_ values (mg/mL).

For the oxidative hemolysis inhibition assay (OxHLIA), sheep erythrocytes were used, as previously described by Lockowandt et al. [43]. Results are expressed as half-maximal inhibitory concentrations (IC_50_ values, μg/mL) calculated for a Δ*t* of 60 min. Trolox (6-hydroxy-2,5,7,8-tetramethylchroman-2-carboxylic acid, purchased from Sigma), was used as a positive control.

#### 2.4.2. Evaluation of Antibacterial Activity

The extracts were tested against five Gram-negative bacteria, namely, *Enterobacter cloacae* (ATCC 49741), *Escherichia coli* (ATCC 25922), *Pseudomonas aeruginosa* (ATCC 9027), *Salmonella enterica* subsp. *enterica* serovar Enteritidis (ATCC 13076), and *Yersinia enterocolitica* (ATCC 8610), and three Gram-positive bacteria, namely, *Bacillus cereus* (ATCC 11778), *Listeria monocytogenes* (ATCC 19111), and *Staphylococcus aureus* (ATCC 25923). The minimum inhibitory (MIC) and minimum bactericidal concentrations were determined for all bacteria using colorimetric assays, following Pires et al. [44]. The MIC was defined as the lowest concentration inhibiting visible bacterial growth, determined by a change from yellow to pink coloration if the microorganisms are viable. The MBC was defined as the lowest concentration required to kill bacteria.

To evaluate the antifungal activity, the methodology described by Heleno et al. [45], using *Aspergillus fumigatus* (ATCC 204305) and *Aspergillus brasiliensis* (ATCC 16404), was used. The organisms were obtained from Frilabo, Porto, Portugal. The minimum inhibitory concentration (MIC) and minimum fungicidal concentration (MFC) were determined by a serial dilution technique using 96-well microplates. The lowest concentrations without visible growth (at the binocular microscope) were defined as the MICs. The lowest concentration with no visible growth was defined as the MFC, indicating 99.5% killing of the original inoculum. The commercial fungicide ketoconazole (Frilabo, Porto, Portugal) was used as positive control.

### 2.5. Statistical Analysis

Three independent samples per mushroom species were analyzed, and the data are expressed as the mean ± standard deviation. All statistical tests were performed at a 5% significance level in RStudio (version 1.1.485—© 2009–2022 RStudio, Inc.) [46]. The homogeneity of variance and normal distribution of the residuals were tested by means of the Shapiro–Wilk and Levene tests, respectively, to fulfill the one-way ANOVA requirements. All dependent variables were compared using Tukey’s tests. When normality or heteroscedasticity could not be verified, the variables were Box–Cox transformed before performing the ANOVA. Kruskal–Wallis tests were carried out when a normal distribution and heteroscedasticity were not achieved after Box–Cox transformation.

## 3. Results and Discussion

The obtained chemical compositions and energetic values are shown in Table 1. Protein contents varied between 3.20 g/100 g dw in *F. antarctica* and 36.60 g/100 g dw in *L. perlatum*. The top-five values concerning of highest protein content was *L. perlatum* (36.60 g/100 g dw), *L. nuda* (30.30 g/100 g dw), *H. dusenii* (22.20 g/100 g dw), *R. patagonica* (18.10 g/100 g dw), and *C. magellanicus* (14.40 g/100 g dw). Comparing with previous studies, *Ramaria patagonica* and *R. botrytis* showed similar results than those reported by other authors with a value of 19.68 g/100 g dw [25] and 16.60 g/100 g dw [14]. However, in a Portuguese mushroom study [16], *R. botrytis* showed higher protein values (39.8 g/100 g dw) than our results. *Flammulina velutipes* (17.89 g/100 g dw) and *C. aegerita* (19.65 g/100 g dw) showed lower levels than those reported by Jacinto-Azevedo et al. [14]. Other studies on *G. gargal* showed similar results with values of 5.96 [25] and 5.00 g/100 g dw [38]. Previous studies on *Cyttaria* have reported higher values than those reported here; for example, *C. espinosae* had values of 17.46 g/100 g dw [14] and *C. darwini* values of 17.20 g/100 g dw [38,47]. However, *C. hariotii* showed similar results (3.35 g/100 g dw) than those reported by Toledo et al. [25]. The protein values for *A. vittelinus, C. magellanicus, F. antarctica*, and *F. endoxantha* are in concordance with other reports [25]. On a dry weight basis, mushrooms normally contain 19 to 35% protein. Therefore, regarding the amount of crude protein, mushrooms are positioned below most animal meats but well above most other foods, including milk, rice, and wheat [48].

The fat content ranged from 0.70 g/100 g dw (*F. antarctica*) to 5.90 g/100 g dw (*G. gargal*). High values were also observed in *A. vitellinus* (4.70 g/100 g dw) and *C. magellanicus* (4.40 g/100 g dw). Furthermore, the crude fat content of *C. aegerita* (1.05 g/100 g dw) was higher than those obtained by Jacinto-Azevedo et al. [14], while *F. velutipes* (1.70 g/100 g dw) was similar as that reported by other authors [14,38]. In addition, levels in *A. vitellinus* (3.49 g/100 g dw)*, C. magellanicus* (2.75 g/100 g dw)*, C. hariotii* (1.31 g/100 g dw)*, G. gargal* (1.79 g/100 g dw), and *L. nuda* (0.84 g/100 g dw) were higher than those obtained by Toledo et al. [25], but were lower in *F. antarctica* (0.83 g/100 g dw)*, F. endoxantha* (1.19 g/100 g dw)*, H. dusenii* (4.29 g/100 g dw), and *R. patagonica* (2.51 g/100 g dw) compared to these authors’ findings. The fat content in *C. hariotii* was similar (2.10 g/100 g dw) than previous reports [47]. Low fat contents in edible mushrooms are one of the reasons why they are recognized as healthy food sources. Chang and Miles [48] have reported fat contents varying between 1 to 15% per 100 g of dried weight, including all types of lipids.

Ash varied from 4.80 g/100 g in *P. ostreatus* to 32.00 g/100 g in *C. magellanicus*. *Cyttaria hariotii* yielded lower values (7.0 g/100 g dw) than what Schmeda-Hirschmann et al. [47] previously reported, and similar values than those reported by Jacinto-Azevedo et al. [14] for *C. espinosae* (4.90 g/100 g dw). However, lower values were observed in *F. velutipes, C. aegerita*, and *R. botrytis* [14], in *G. gargal* [47], and in *F. antarctica* and *F. endoxantha* [25]. Similar results to Toledo et al. [25] were found for *L. nuda* (8.58 g/100 g dw) and *R. patagonica* (8.47 g/100 g dw), but lower for *L. nuda* (18.5 g/100 g dw) compared to Barros et al. [11]. The ash content in edible mushrooms ranges from 1 to 29 g/100 g dry matter and comprise a source of essential minerals. Concentrations of P, K, Ca, Na, and Mg constitute more than 56% of the total ash content [48].

Carbohydrates represent the most abundant nutrient, varying between 49.00 g/100 g dw (*C. magellanicus*) and 89.70 g/100 g dw (*F. antarctica*), closely followed by *P. ostreatus* and C. *hariotii*. Previous reports showed similar results for *F. endoxantha*, *G. gargal*, and *R. botrytis*, lower for *A. vitellinus*, *C. magellanicus*, *C. aegerita*, *C. hariotii*, *F. velutipes*, *H. dusenii*, and *L. nuda*, and higher for *P. ostreatus* and *R. patagonica* [14,25].

Energetic values ranged from 293.00 Kcal/100 g dw in *C. magellanicus* to 398.00 Kcal/100 g dw in *G. gargal*. Other species with high energetic values were *C. hariotii* (392.00 kcal/100 g dw), *A. vitellinus* (387.00 kcal/100 g dw), and *P. ostreatus* (386.00 kcal/100 g dw).

Concerning sugar composition (Table 2), mannitol and trehalose were the principal sugars, which is in agreement with the data presented in the literature; they are essential in energetic metabolism and necessary in the synthesis of storage or structural polysaccharides [38]. Mannitol content was significantly higher for *A. vitellinus* (8.83 g/100 g dw), *R. botrytis* (6.34 g/100 g dw), and *R. patagonica* (8.64 g/100 g dw), although absent in *F. endoxantha*, in concordance with Toledo et al. [25]. The support and expansion of the mushroom fruiting bodies is guaranteed by the presence of mannitol [49]. Trehalose predominated in *C. xiphidipus* (17.60 g/100 g dw), *P. ostreatus* (15.81 g/100 g dw), *C. aegerita* (13.54 g/100 g dw), and *G. sordulenta* (11.25 g/100 g dw), but was absent in *R. botrytis* and *R. patagonica*. The ingestion, hydrolysis, absorption, and metabolism of trehalose is highly similar to all the other digestible disaccharides [49]. On the other hand, fructose and one unidentified sugar were predominant in all three *Fistulina* species *(F. endoxantha, F. antarctica,* and *F. pumiliae*) in concordance with previous reports [25]. Fructose was also the predominant sugar in *Flammulina velutipes* (8.56 g/100 g dw), agreeing with Reis et al. [50], while it was absent in *A. vitellinus*, *C. magellanicus*, *L. perlatum*, and *P. ostreatus*, and present in lower abundance in the rest of the species. In terms of total sugar content, *F. endoxantha* revealed the highest value (33.88 g/100 g dw), while *G. gargal* the lowest (3.63 g/100 g dw). This is the first study that reports the composition in free sugars of endemic species *C. xiphidipus*, *F. pumiliae*, and *G. sordulenta*.

Organic acids comprise a group of mono-, di-, and tricarboxylic acids physiologically occurring as intermediates in a variety of intracellular metabolic pathways, such as catabolism of amino acids, the tricarboxylic acid cycle, and neurotransmitters, as well as in cholesterol biosynthesis [51]. The organic acid composition is presented in Table 2. Most of the analyzed specimens had oxalic, malic, and fumaric acids in their composition. On the other hand, quinic, shikimic, citric, and succinic acids were present just in a few species. Citric acid was detected in high amounts in *C. magellanicus* (57.40 mg/100 g dw), *R. patagonica* (57.31 mg/100 g dw), *A. vitellinus* (31.11 mg/100 g dw), and *C. hariotii* (23.41 mg/100 g dw). The oxalic acid content was significantly higher for *L. nuda* (66.72 mg/100 g dw). Shikimic acid was present just in *C. aegerita* (0.72 mg/100 g dw) and *P. ostreatus* (1.07 g/100 mg dw). Quinic acid was only present in *C. xiphidipus* (541.57 mg/100 g dw), *F. antarctica* (0.54 mg/100 g dw), and *L. nuda* (487.50 mg/100 g dw). High amounts of succinic acid were detected in *A. vitellinus* (452.79 mg/100 g dw) and *L. nuda* (277.39 mg/100 g dw). *Aleurodiscus vitellinus* and *C. magellanicus* showed the most optimum amount of malic acid (74.58 mg/100 g dw) and fumaric acid (11.02 mg/100 g dw), respectively. Ascorbic acid was not detected. Analyzing the total organic acids profile, the highest value was revealed by *L. nuda* (832.92 mg/100 g dw), while the lowest by *H. dusenii* (5.13 mg/100 g dw).

Table 3 presents the fatty acid composition of each species, along with the values of the total saturated fatty acids (SFAs), polyunsaturated fatty acids (PUFAs), and monounsaturated fatty acids (MUFAs). Linoleic acid (C18:2), oleic acid (C18:1), and palmitic acid (C16:0) were the major fatty acid found in the studied species, in concordance with different studies [10,25]. In total, 27 fatty acids were identified and quantified. Due to the high contribution of linoleic acid, PUFAs were the main group of fatty acids in *C. xiphidipus* (43%), *C. aegerita* (48.70%), *C. hariotii* (48.40%), *F. antarctica* (48.40%), *L. nuda* (54.30%), and *L. perlatum* (68%). MUFAs were the main group of fatty acids in *A. vitellinus* (54.59%), *G. gargal* (58.9%), *H. dusenii* (52.60%), *R. botrytis* (43.91%), and *R. patagonica* (39.37%), due to the high contribution of oleic acid, in concordance with Toledo et al. [25]. SFAs predominate in *C. magellanicus*, *F. endoxantha*, *F. pumiliae*, *F. velutipes*, *G. sordulenta*, and *P. ostreatus* due to the palmitic acid content.

Regarding the favorable effect of fatty acids on human health, oleic acid, a monounsaturated fatty acid ω-9 series, also present in vegetable oils, is known for its efficiency in reducing cholesterol levels, preventing cardiovascular diseases [52,53].

Within polyunsaturated fatty acids, ω-3 and ω-6 are the most abundant in mammals. Its precursors, α-linolenic acid (ALA) and linoleic acid (LA), are considered essential fatty acids, as the body requires them for normal operation but which cannot be synthesized endogenously [54]. Within the series of omega-3, the most important in the human diet are eicosapentaenoic acid (EPA) and docosahexaenoic acid (DHA), both difficult to synthesize endogenously and with important functions in the human body. DHA is a structural fatty acid, since it forms part of cell membranes and is also important for visual (it makes up 20% of all fatty acids present in the retina) and neuronal development during gestation and early childhood [55,56]. In our study, *C. hariotii* showed high amounts of DHA (7.74%). Interestingly, some ethnomycological reports [57,58] showed that in the Selknam, Ahonikenk, and Kawesqar native Patagonian populations, after giving birth and while the quarantine lasted, the mothers lived exclusively on *Cyttaria* fungus (*C. darwinii* and *C. hariotii*). In the omega-6 series one has to pay special attention to γ-linolenic acid (GLA) and arachidonic acid (AA), important for prostaglandin production and anti-inflammatory activity [55]. From all the above, the importance of supplementing a diet with fatty acids, which are clearly present in edible mushrooms, can be deduced, since their current intake is insufficient.

The ergosterol content (Table 4) ranged from 0.40 mg/100 g in *C. hariotii* to 123.57 mg/100 g dw in *G. gargal*. In other studies of edible and medicinal mushrooms, similar differences have been reported, ranging from traces for *Armilaria mellea*, to values of 25.71 mg/100 g dw for *Laetiporus sulphureus* or 445.32 mg/100 g dw for *Macrolepiota procera*. The differences in ergosterol and other nutritional and bioactive compounds depend on the species, stage of development, tissues, nutrient substrate, and microclimate [26,59]. Vieira Junior et al. [39] showed that ergosterol biosynthesis and its bioconversion into ergocalciferol were also affected by the cultivation process in *Agaricus subrufescens* production, showing differences between field culture and controlled conditions. Ergosterol is metabolized into a prohormone—vitamin D. Its action is related with bone mineral metabolism and with the balance of phosphorus and calcium, related to various mechanisms such as secretion and effect of insulin, regulation of the renin–angiotensin–aldosterone system, endothelial function, cell cycle control and apoptosis, immunological self-tolerance, and immune response against infections, among other effects [60]. For these reason, edible mushrooms are promising sources of vitamin D, and thus able to improve food supplements for human consumption.

Regarding phenolic compounds (Table 5), four phenolic acids (gallic, *p*-hydroxybenzoic, protocatechuic, and *p*-coumaric) and two related compounds (3-(3,4-dihydroxyphenyl)-lactic acid and gallic acid monohydrate) were identified and quantified. All the studied species presented gallic acid between 0.80 (for *C. hariotii*, consistent with previous reports [25]) and 7.36 mg/g dw (*C. xiphidipus*). The *p*-hydroxybenzoic acid was present in *C. magellanicus*, *C. xiphidipus*, *C. aegerita*, *F. endoxantha*, *F. pumiliae*, *L. nuda*, *L. perlatum*, and *P. ostreatus*, with values of 0.55 mg/g dw for *F. pumiliae* and 40.33 mg/g dw for *L. perlatum*. Protocatechuic acid was present in *C. magellanicus*, *C. xiphidipus*, *C. aegerita*, *F. antarctica*, *F. endoxantha*, *F. pumiliae*, *G. gargal*, *R. botrytis*, and *R. patagonica*, with the highest values for *F. endoxantha* (7.65 mg/g dw) and *R. patagonica* (5.30 mg/g dw). In addition, *p*-coumaric acid was registered only in *C. aegerita*, *L. nuda*, and *L. perlatum.* In comparison with the other species, *L. perlatum* presented a significantly higher value of total phenolic acids (51.40 mg/g dw), attributable to the proportion of *p*-hydroxybenzoic acid. In the study of Toledo et al. [25], *L. nuda* did not present phenolic compounds; however, in concordance with Barros et al. [61], in this study, *L. nuda* presented *p*-coumaric, gallic, and *p*-hydroxybenzoic acids.

Table 6 shows the in vitro antioxidant activity of the studied species. *Ramaria patagonica* (156 µg/mL), *R. botrytis* (167 µg/mL), *G. sordulenta* (299 µg/mL), and *A. vitellinus* (551 µg/mL) presented the best results in the TBARS assays; meanwhile, *L. perlatum* (90 µg/mL), *L. nuda* (93 µg/mL), *A. vitellinus* (113 µg/mL), and *G. sordulenta* (155 µg/mL) presented the best results in OxHLIA, all with IC_50_ values ≤ 1000 µg/mL. The result of the antioxidant activity in OxHLIA for *L. perlatum* is in concordance with its highest total levels of phenolic compounds. That the antioxidant activity of mushrooms correlates with the phenolic compounds content has already been reported [62]. *Fistulina antarctica* comparatively presented the lowest antioxidant activity (IC_50_ of 2627 µg/mL for TBARS and of 1066 µg/mL for OxHLIA), in concordance with Toledo et al. [25]. *Lepista nuda* showed lower values (711 µg/mL) for the TBARS assay compared to those reported by other authors, with IC_50_ values of 5800 µg/mL [11] and 6100 µg/mL [25]. The high antioxidant activity of *Ramaria* is in agreement with the literature; for example, for *R. flava, R. botrytis*, and *R. subaurantiaca* with DPPH assays by Jacinto-Azevedo et al. [14], and for *R. patagonica* with different assays by Toledo et al. [25]. The antioxidant activity by DPPH or reducing the power test was also evaluated applying different extracting methodologies in *Grifola* samples; for example, Brujin et al. [63,64], using different solvents or heat treatments in *Grifola gargal*, and Postemsky et al. [65], using wheat grain biotransformed with mycelium of *G. gargal* and *G. sordulenta*. Among the three edible *Rusulla* species, *R. integra* ethanolic extract showed the best antihemolytic activity, with an IC_50_ value of 139 ± 3 μg/mL, also for a 60 min Δ*t* [25]. This is the first study on the anti-hemolytic capacity of wild edible species.

All extracts were tested against ten bacteria and fungi considered food contaminants (Table 7). Each mushroom species showed different intensities of positive antimicrobial activity against the tested microorganisms. The antibacterial effects were more effective against *Salmonella enterocolitica*, *Yersinia enterocolitica* (Gram-negative bacteria), and *Staphylococcus aureus* (Gram-positive bacteria). The antifungal effect was more effective in *Aspergillus brasiliensis.* Among these active extracts, those produced by *A. vitellinus* (MIC 1.25 mg/mL) *F. velutipes* (MIC 2.5 mg/mL), *G. gargal* (MIC 2.5 mg/mL), *P. ostreatus* (MIC 2.5 mg/mL), and *R. botrytis* (MIC 1.25 mg/mL) exhibited a good inhibitory activity against *Yersinia enterocolitica*; and the extracts from *G. sordulenta* and *R. botrytis* (MIC 0.3 mg/mL) against *Staphylococcus aureus*. The Gram-negative bacteria, *Enterobacter cloacae, Escherichia coli*, and *Pseudomonas aeruginosa*, and the Gram-positive bacteria *Bacillus cereus* and *Listeria monocytogenes* were less sensitive to the extracts used. *C. hariotii* showed no activity against the analyzed bacteria. None of the extracts presented bactericidal and fungicidal activity.

In addition, *A. vitellinus*, *C. magellanicus*, *C. xiphidipus*, *F. endoxantha*, *H. dusenii*, and *R. patagonica* had fungistatic effects (MIC 0.15 mg/mL) against *A. brasiliensis* and *A. fumigatus*.

All these results must be considered taking into account the already established fact that the chemical composition of mushrooms could vary with the genetic structure and strains within the same species. Our ranks also considered dehydrated, complete fruiting bodies, in the mature stage, with no stratification by site conditions nor post-harvest treatments. Maturation stage at harvest, a specific part of the mushroom analyzed (stem, cup, lamellae), and environmental variables, such as soil composition, as well as the postharvest preservation method (freeze dry, oven-dry, cooled, fresh) and cooking process may affect their chemical composition [3].

## 4. Conclusions

This study highlights the value of the native and endemic mushrooms of the Patagonian forest, regarding, for example, their antioxidant qualities, as in the case of *Ramaria* spp., or their energetic value, as in the case of *G. gargal* and *C. hariotti*. Species such as *C. aegerita*, *F. velutipes*, *L. nuda*, and *P. ostreatus* demonstrated the importance of edible mushroom with a cosmopolitan distribution growing in native forests, resulting in an invaluable source of food with high protein values, low contents of fat, along with other bioactive compounds with remarkable antioxidant and antimicrobial activity. The data provided by this study, along with previous ones, will strengthen and support the inclusion of new species of wild edible fungi in the Argentine food code. In this way, we expect to revalue these resources as non-timber forest products from Patagonia, promoting multiple and sustainable uses of native forests.

## Figures and Tables

**Figure 1 foods-11-03516-f001:**
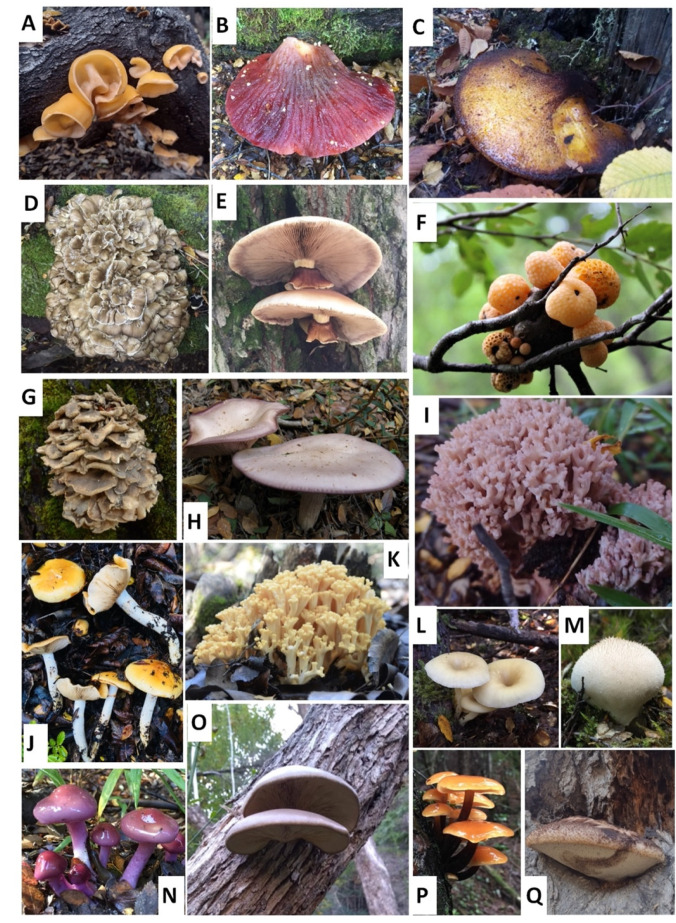
Samples of the wild studied mushrooms. (**A**). *Aleurodiscus vitellinus*; (**B**). *Fistulina antarctica*; (**C**). *Fistulina endoxantha*; (**D**). *Grifola sordulenta*; (**E**). *Cyclocybe aegerita*; (**F**). *Cyttaria hariotii*; (**G**). *Grifola gargal*; (**H**). *Lepista nuda*; (**I**). *Ramaria botrytis*; (**J**). *Cortinarius xiphidipus*; (**K**). *Ramaria patagonica*; (**L**). *Hydropus dusenii*; (**M**). *Lycoperdon perlatum*; (**N**). *Cortinarius magellanicus*; (**O**). *Pleurotus ostreatus*; (**P**). *Flammulina velutipes*; (**Q**). *Fistulina pumiliae*.

**Table 1 foods-11-03516-t001:** Proximate composition (g/100 g) and energetic value (kcal/100 g) of the studied wild mushrooms (mean ± SD). For each mushroom sample, means within a column with different letters differ significantly (*p* < 0.05).

Species	Total Fat	Crude Protein	Carbohydrates	Ash	Energy
*A. vitellinus*	4.70 ± 0.20^b^	5.26 ± 0.01^kl^	81.00 ± 1.00^c^	8.90 ± 0.30^ef^	387.74 ± 0.04^bc^
*C. magellanicus*	4.40 ± 0.20^b^	14.40 ± 0.10^e^	49.00 ± 2.00^i^	32.00 ± 1.00^a^	293.00 ± 4.00^i^
*C. xiphidipus*	2.01 ± 0.04^e^	12.30 ± 0.40^fg^	74.80 ± 0.10^d^	10.9 ± 0.30^d^	366.00 ± 1.00^f^
*C. aegerita*	2.60 ± 0.20^d^	10.70 ± 0.30^h^	63.00 ± 1.00^f^	23 ± 1.00^b^	320.00 ± 5.00^h^
*C. hariotii*	2.60 ± 0.20^d^	5.87 ± 0.07^jk^	86.20 ± 0.20^b^	5.3 ± 0.30^h^	392.00 ± 2.00^ab^
*F. antarctica*	0.70 ± 0.02^f^	3.20 ± 0.10^m^	89.70 ± 0.20^a^	6.4 ± 0.30^g^	378.00 ± 1.00^de^
*F. endoxantha*	0.77 ± 0.04^f^	4.70 ± 0.20^l^	80.00 ± 1.00 ^c^	15 ± 1.00^c^	345.00 ± 2.00^g^
*F. pumiliae*	1.02 ± 0.02^f^	6.53 ± 0.04^j^	82.00 ± 1.00^c^	10 ± 1.00^de^	364.00 ± 2.00^f^
*F. velutipes*	1.70 ± 0.01^e^	11.40 ± 0.30^gh^	66.00 ± 1.00^f^	21.20 ± 0.40^b^	324.00 ± 2.00^h^
*G. gargal*	5.90 ± 0.30^a^	5.20 ± 0.10^kl^	81.10 ± 0.10^c^	7.90 ± 0.40^f^	398.00 ± 3.00^a^
*G. sordulenta*	1.90 ± 0.10^e^	12.60 ± 0.10^f^	71.00 ± 1.00^e^	14.00 ± 1.00^c^	353.00 ± 3.00^g^
*H. dusenii*	3.10 ± 0.10^cd^	22.00 ± 1.00^c^	63.00 ± 1.00^f^	11.30 ± 0.30^d^	370.40 ± 0.40^ef^
*L. nuda*	3.30 ± 0.20^c^	30.30 ± 0.10^b^	57.60 ± 0.20^g^	8.70 ± 0.40^f^	382.00 ± 3.00^cd^
*L. perlatum*	3.32 ± 0.10^c^	36.60 ± 0.10^a^	52.00 ± 0.30^h^	8.10 ± 0.40^f^	384.00 ± 2.00^bcd^
*P. ostreatus*	1.05 ± 0.05^f^	7.65 ± 0.01^i^	86.53 ± 0.01^b^	4.80 ± 0.10^h^	386.00 ± 1.00^bcd^
*R. botrytis*	3.40 ± 0.10^c^	12.60 ± 0.40^f^	73.00 ± 1.00^de^	11.00 ± 1.00^d^	373.00 ± 1.00^ef^
*R. patagonica*	0.90 ± 0.02^f^	18.10 ± 0.20^d^	72.40 ± 0.30^de^	8.60 ± 0.10^f^	370.00 ± 1.00^ef^

**Table 2 foods-11-03516-t002:** Organic acid and sugar composition (mg/100 g) of the studied wild mushrooms (mean ± SD). For each mushroom sample, means within a column with different letters differ significantly (*p* < 0.05).

Organic Acids								Sugars			
Species	Oxalic	Quinic	Malic	Shikimic	Citric	Succinic	Fumaric	Not Identified	Fructose	Mannitol	Trehalose
*A. vitellinus*	nd	nd	74.58 ± 1.03^a^	nd	31.10 ± 0.40^b^	452.00 ± 2.00^a^	0.05 ± 0.01^n^	0.17 ± 0.01^i^	nd	8.80 ± 0.10^a^	3.16 ± 0.03^hi^
*C. magellanicus*	2.02 ± 0.05^fg^	nd	10.90 ± 0.30^f^	nd	57.40 ± 0.03^a^	8.10 ± 0.20^d^	11.02 ± 0.02^a^	0.15 ± 0.01^i^	nd	4.80 ± 0.20^c^	3.04 ± 0.12^hi^
*C. xiphidipus*	0.93 ± 0.01^gh^	542.00 ± 3.00^a^	nd	nd	nd	nd	3.93 ± 0.01^c^	nd	0.70 ± 0.01^e^	3.30 ± 0.10^d^	17.60 ± 0.40^a^
*C. aegerita*	0.49 ± 0.01^jk^	nd	17.29 ± 0.04^d^	0.72 ± 0.04^b^	nd	nd	3.40 ± 0.02^e^	nd	0.31 ± 0.02^f^	1.22 ± 0.04^g^	13.54 ± 0.02^c^
*C. hariotii*	0.09 ± 0.01^m^	nd	13.40 ± 0.30^e^	nd	23.40 ± 0.20^a^	nd	2.39 ± 0.01^i^	3.19 ± 0.02^d^	2.10 ± 0.10^c^	0.33 ± 0.01^j^	2.10 ± 0.10^j^
*F. antarctica*	0.05 ± 0.02^m^	0.50 ± 0.10^b^	13.60 ± 0.30^e^	nd	nd	nd	2.54 ± 0.01^h^	12.00 ± 1.00^a^	10 ± 1.00^b^	1.00 ± 0.10^h^	8.00 ± 1.00^e^
*F. endoxantha*	0.35 ± 0.01^lm^	nd	16.80 ± 1.10^d^	nd	nd	nd	5.34 ± 0.04^b^	7.00 ± 0.40^b^	23.18 ± 1.02^a^	nd	3.70 ± 0.10^h^
*F. pumiliae*	0.44 ± 0.01^kl^	nd	35.00 ± 1.00^b^	nd	nd	nd	5.33 ± 0.01^b^	6.60 ± 0.30^b^	9.10 ± 0.40^b^	3.40 ± 0.10^d^	5.30 ± 0.20^fg^
*F. velutipes*	3.90 ± 0.10^cd^	nd	32.60 ± 0.50^b^	nd	nd	nd	5.40 ± 0.02^b^	1.87 ± 0.01^e^	8.60 ± 0.10^b^	1.64 ± 0.02^f^	4.70 ± 0.40^g^
*G. gargal*	2.40 ± 0.10^ef^	nd	1.90 ± 0.02^h^	nd	nd	nd	0.34 ± 0.01^m^	0.16 ± 0.01^i^	0.77 ± 0.04^d^	0.25 ± 0.01^k^	2.50 ± 0.10^ij^
*G. sordulenta*	0.72 ± 0.02^hi^	nd	21.40 ± 0.30^c^	nd	nd	nd	2.79 ± 0.02^g^	nd	0.30 ± 0.01^f^	0.50 ± 0.02^i^	11.30 ± 0.20^d^
*H. dusenii*	3.39 ± 0.03^de^	nd	nd	nd	nd	nd	1.74 ± 0.01^j^	0.26 ± 0.02^h^	0.18 ± 0.01^g^	2.30 ± 0.10^e^	5.70 ± 0.10^f^
*L. nuda*	66.70 ± 1.90^a^	488.00 ± 38.00^a^	nd	nd	nd	277.00 ± 2.00^b^	1.31 ± 0.01^k^	0.57 ± 0.01^g^	0.17 ± 0.01^g^	1.29 ± 0.01^g^	7.10 ± 0.10^e^
*L. perlatum*	4.90 ± 0.10^def^	nd	8.70 ± 0.20^g^	nd	nd	87.00 ± 3.00^c^	1.20 ± 0.10^l^	3.88 ± 0.03^c^	nd	nd	5.50 ± 0.10^fg^
*P. ostreatus*	0.59 ± 0.03^ij^	nd	11.90 ± 0.30^ef^	1.07 ± 0.03^a^	nd	83.00 ± 1.00^c^	2.54 ± 0.01^h^	nd	nd	3.30 ± 0.10^d^	15.80 ± 0.40^b^
*R. botrytis*	25.36 ± 0.02^b^	nd	nd	nd	nd	nd	3.02 ± 0.02^f^	1.59 ± 0.05^f^	0.76 ± 0.04^de^	6.34 ± 0.04^b^	Nd
*R. patagonica*	25.60 ± 0.10^ab^	nd	nd	nd	57.30 ± 0.80^c^	nd	3.67 ± 0.01^d^	nd	0.86 ± 0.04^d^	8.60 ± 0.40^a^	Nd

**Table 3 foods-11-03516-t003:** Fatty acid composition (%) of the studied wild mushrooms (mean ± SD). (C8:0) caprylic acid; (C10:0) capric acid; (C11:0) undecanoic acid; (C12:0) lauric acid; (C13:0) tridecanoic acid; (C14:0) myristic acid; (C15:0) pentadecanoic acid; (C16:0) palmitic acid; (C16:1) palmitoleic acid; (C17:0) heptadecanoic acid; (C18:0) stearic acid; (C18:1n9) oleic acid; (C18:2n6) linoleic acid; (C18:3n3) α-linolenic acid; (C20:0) arachidic acid; (C20:1) cis-11-eicosaenoic acid; (C20:2) cis-11,14-eicosadienoic acid; (C20:4n6) arachidonic acid; (C20:5n3) cis-5,8,11,14,17-eicosapentaenoic acid; (C21:0) heneicosanoic acid; (C22:0) behenic acid; (C22:1) erucic acid; (C22:2) Cis-13,16-docosadienoic acid; (C23:0) tricosanoic acid; (C24:0) lignoceric acid; (C24:1) nervonic acid; (C22:6n3) docosahexaenoic acid; (SFAs) saturated fatty acids; (MUFAs) monounsaturated fatty acids; (PUFAs) polyunsaturated fatty acids. For each mushroom sample, means within a column with different letters differ significantly (*p* < 0.05).

Species	*A. vitellinus*	*C. magellanicus*	*C. xiphidipus*	*C. aegerita*	*C. hariotii*	*F. anta* *rctica*	*F. endoxantha*	*F. pumiliae*	*F. velutipes*
C8:0	nd	0.63 ± 0.02^b^	nd	0.05 ± 0.01^d^	nd	nd	nd	0.81 ± 0.02^a^	nd
C10:0	nd	nd	nd	nd	nd	nd	nd	nd	nd
C11:0	nd	nd	nd	nd	nd	nd	1.50 ± 0.03^a^	0.86 ± 0.04^b^	1.70 ± 0.03^a^
C12:0	nd	0.11 ± 0.01^c^	nd	0.10 ± 0.01^c^	nd	0.28 ± 0.01^b^	nd	nd	nd
C13:0	0.15 ± 0.04^cd^	0.37 ± 0.01^a^	0.18 ± 0.01^bcd^	0.24 ± 0.01^b^	0.13 ± 0.01^d^	nd	nd	nd	nd
C14:0	0.28 ± 0.01^g^	0.35 ± 0.01^f^	0.23 ± 0.01^gh^	0.37 ± 0.01^ef^	0.22 ± 0.01^hi^	1.33 ± 0.02^b^	1.80 ± 0.05^ab^	2.4 ± 0.10^a^	nd
C15:0	0.82 ± 0.01^g^	1.47 ± 0.03^d^	0.50 ± 0.01^i^	0.44 ± 0.01^j^	0.15 ± 0.01^l^	0.71 ± 0.01^g^	1.58 ± 0.04^c^	1.66 ± 0.04^c^	nd
C16:0	21.90 ± 0.1^d^	53.40 ± 0.20^b^	17.00 ± 1.00^f^	19.60 ± 0.10^e^	17.30 ± 0.20^f^	18.30 ± 0.10^e^	51.10 ± 0.70^b^	52.6 ± 0.50^b^	61.00 ± 1.00^a^
C16:1	1.70 ± 0.04^a^	0.61 ± 0.02^d^	0.31 ± 0.01^f^	0.57 ± 0.01^d^	nd	0.63 ± 0.03^d^	1.40 ± 0.01^b^	nd	nd
C17:0	0.16 ± 0.01^h^	nd	0.14 ± 0.01^i^	0.21 ± 0.01^g^	0.19 ± 0.01^g^	nd	3.19 ± 0.05^c^	5.4 ± 0.20^a^	nd
C18:0	4.40 ± 0.01^de^	7.49 ± 0.02^b^	3.2 ± 0.10^h^	5.10 ± 0.10^c^	4.65 ± 0.03^cd^	2.92 ± 0.02^h^	7.10 ± 0.30^b^	nd	nd
C18:1n9t	nd	nd	nd	nd	nd	nd	nd	5.9 ± 0.10^c^	nd
C18:1n9c	52.60 ± 0.10^b^	19.9 ± 0.10^l^	31.30 ± 0.50^g^	20.2 ± 0.20^k^	6.50 ± 0.10^m^	26.80 ± 0.30^j^	32.40 ± 0.30^f^	30.3 ± 0.10^h^	nd
C18:2n6t	0.22 ± 0.01^f^	nd	1.3 ± 0.10^c^	1.75 ± 0.02^b^	0.58 ± 0.01^d^	nd	nd	nd	37.00 ± 10.00 ^a^
C18:2n6c	13.50 ± 0.10^j^	12.3 ± 0.10^k^	40.4 ± 0.30^e^	44.70 ± 0.20^d^	25.10 ± 0.20^i^	48.30 ± 0.20^c^	nd	nd	nd
C18:3n3	0.96 ± 0.01^b^	nd	nd	0.14 ± 0.01^d^	12.50 ± 0.10^a^	nd	nd	nd	nd
C20:0	0.74 ± 0.02^f^	1.10 ± 0.01^bc^	0.80 ± 0.01^de^	0.77 ± 0.01^ef^	2.90 ± 0.10^a^	nd	nd	nd	nd
C20:1	0.22 ±0.01^h^	nd	0.75 ± 0.02^d^	0.67 ± 0.01^e^	0.43 ± 0.01^f^	nd	nd	nd	nd
C20:2	0.73 ± 0.04^d^	nd	0.85 ± 0.02^d^	1.10 ± 0.01^c^	1.50 ± 0.10^b^	nd	nd	nd	nd
C20:4n6	nd	nd	nd	nd	0.44 ± 0.01	nd	nd	nd	nd
C20:5n3	nd	nd	nd	nd	nd	nd	nd	nd	nd
C21:0	nd	nd	nd	0.69 ± 0.01^a^	0.25 ± 0.01^b^	nd	nd	nd	nd
C22:0	0.55 ± 0.01^h^	nd	1.11 ± 0.01^c^	0.79 ± 0.01^f^	7.40 ± 0.10^a^	0.46 ± 0.01^h^	nd	nd	nd
C22:1	nd	1.8 ± 0.10^a^	0.24 ± 0.01^b^	0.29 ± 0.01^b^	0.27 ± 0.01^b^	nd	nd	nd	nd
C22:2	nd	nd	nd	0.33 ± 0.01^c^	0.46 ± 0.01^b^	nd	nd	nd	nd
C23:0	0.19 ± 0.01^d^	0.42 ± 0.01^c^	nd		nd	nd	nd	nd	nd
C24:0	0.47 ± 0.01^f^	nd	0.53 ± 0.01^f^	0.90 ± 0.02^d^	6.75 ± 0.02^a^	0.44 ± 0.01^f^	nd	nd	nd
C24:1	0.07 ± 0.01^g^	nd	0.71 ± 0.02^c^	0.36 ± 0.01^d^	4.47 ± 0.05^a^	nd	nd	nd	nd
C22:6n3	0.27 ± 0.01^d^	nd	0.38 ± 0.01^c^	0.69 ± 0.02^b^	7.74 ± 0.01^a^	nd	nd	nd	nd
SFAs	29.70 ± 0.10^gh^	65.40 ± 0.20^b^	23.70 ± 0.40^i^	29.20 ± 0.01^h^	39.90 ± 0.20^ef^	24.40 ± 0.05^i^	66.20 ± 0.30^a^	63.80 ± 0.20^c^	63.00 ± 1.00^cd^
MUFAs	54.60 ± 0.10^ab^	22.40 ± 0.10^i^	33.00 ± 1.00^g^	22.10 ± 0.20^i^	11.70 ± 0.10^j^	27.30 ± 0.20^h^	33.80 ± 0.30^g^	36.20 ± 0.20^ef^	nd
PUFAs	15.73 ± 0.01^i^	12.30 ± 0.10^j^	43.00 ± 0.20^d^	48.70 ± 0.20^c^	48.40 ± 0.20^c^	48.30 ± 0.20^c^	nd	nd	37.00 ± 1.00^f^
**Species**		* **G. gargal** *	* **G. sordulenta** *	* **H. dusenii** *	* **L. nuda** *	* **L. perlatum** *	* **P. ostreatus** *	* **R. botrytis** *	* **R. patagonica** *
C8:0		nd	nd	0.25 ± 0.01^c^	Nd	nd	nd	nd	nd
C10:0		nd	nd	Nd	nd	nd	0.40 ± 0.02	nd	nd
C11:0		nd	nd	Nd	nd	nd	0.84 ± 0.01^b^	nd	0.17 ± 0.01^c^
C12:0		nd	0.23 ± 0.01^b^	0.11 ± 0.01^c^	nd	nd	0.37 ± 0.01^a^	nd	nd
C13:0		nd	0.20 ± 0.01^bc^	0.15 ± 0.01^cd^	nd	nd	0.41 ± 0.02^a^	nd	nd
C14:0		nd	1.20 ± 0.04^bc^	0.36 ± 0.01^f^	0.36 ± 0.01^f^	0.47 ± 0.01^de^	0.77 ± 0.02^cd^	nd	0.17 ± 0.01^i^
C15:0		nd	3.23 ± 0.03^b^	1.12 ± 0.01^e^	0.30 ± 0.01^k^	nd	5.40 ± 0.10^a^	0.86 ± 0.01^f^	0.60 ± 0.01^h^
C16:0		20.00 ± 1.00^e^	21.33 ± 0.03^d^	27.30 ± 0.1^c^	15.40 ± 0.20^g^	17.00 ± 1.00^f^	53.00 ± 1.00^b^	9.67 ± 0.02^i^	13.90 ± 0.10^h^
C16:1		0.63 ± 0.01^d^	nd	Nd	0.74 ± 0.02^c^	nd	nd	nd	0.43 ± 0.01^e^
C17:0		nd	1.46 ± 0.04^d^	0.49 ± 0.02^f^	nd	nd	4.20 ± 0.20^b^	1.50 ± 0.02^d^	0.77 ± 0.01^e^
C17:1		nd	nd	Nd	nd	nd	nd	0.30 ± 0.01^a^	0.29 ± 0.01^a^
C18:0		8.90 ± 0.10^a^	3.94 ± 0.05^fg^	9.32 ± 0.03^a^	2.05 ± 0.05^i^	3.19 ± 0.05^h^	nd	4.01 ± 0.01^ef^	3.70 ± 0.10^g^
C18:1n9t		nd	nd	Nd	nd	11.50 ± 0.10^a^	7.43 ± 0.02^b^	nd	nd
C18:1n9c		58.20 ± 0.20^a^	28.90 ± 0.10^i^	46.80 ± 0.03^c^	25.50 ± 0.10^j^	nd	27.00 ± 1.00^j^	42.67 ± 0.01^d^	37.60 ± 0.10^e^
C18:2n6t		nd	0.39 ± 0.02^e^	Nd	nd	nd	nd	0.19 ± 0.01^g^	nd
C18:2n6c		11.10 ± 0.30^l^	31.45 ± 0.01^h^	1.53 ± 0.04^m^	54.30 ± 0.10^b^	68.00 ± 1.00^a^	nd	35.04 ± 0.01^g^	38.14 ± 0.02^f^
C18:3n3		0.50 ± 0.02^c^	nd	Nd	nd	nd	nd	nd	nd
C20:0		nd	2.92 ± 0.01^a^	1.78 ± 0.01^b^	nd	nd	nd	0.89 ± 0.01^cd^	nd
C20:1		nd	1.50 ± 0.03^b^	3.48 ± 0.04^a^	nd	nd	nd	0.33 ± 0.01^g^	0.88 ± 0.04^c^
C20:2		nd	nd	1.76 ± 0.01^a^	nd	nd	nd	nd	nd
C20:4n6		nd	nd	Nd	nd	nd	nd	nd	nd
C20:5n3		nd	nd	Nd	nd	nd	nd	0.29 ± 0.01	nd
C21:0		nd	nd	Nd	nd	nd	nd	nd	nd
C22:0		0.33 ± 0.01^i^	0.97 ± 0.01^d^	0.73 ± 0.01^g^	0.55 ± 0.01^h^	nd	nd	1.56 ± 0.01^b^	0.95 ± 0.02^e^
C22:1		nd	nd	Nd	nd	nd	nd	0.48 ± 0.01^ab^	nd
C22:2		nd	1.15 ± 0.04^a^	Nd	nd	nd	nd	0.25 ± 0.01^d^	nd
C23:0		nd	nd	0.28 ± 0.01^d^	nd	nd	nd	0.78 ± 0.01^b^	1.15 ± 0.01^a^
C24:0		0.51 ± 0.02^f^	0.99 ± 0.01^cd^	1.62 ± 0.01^b^	0.50 ± 0.02^f^	nd	nd	0.71 ± 0.02^e^	1.03 ± 0.02^c^
C24:1		nd	nd	2.32 ± 0.01^b^	0.33 ± 0.01^d^	nd	nd	0.14 ± 0.01^f^	0.18 ± 0.01^e^
C22:6n3		0.12 ± 0.01^f^	0.14 ± 0.01^e^	0.65 ± 0.01^b^	nd	nd	nd	0.34 ± 0.01^c^	0.08 ± 0.02^g^
SFA		29.40 ± 0.40^h^	36.47 ± 0.03^fg^	43.47 ± 0.04^de^	19.10 ± 0.20^k^	20.66 ± 1.00 ^j^	66.00 ± 1.00^b^	19.98 ± 0.01^j^	22.42 ± 0.02^i^
MUFA		58.90 ± 0.20^a^	30.40 ± 0.10^h^	52.60 ± 0.01^bc^	26.60 ± 0.10^h^	11.50 ± 0.10^j^	34.00 ± 1.00^f^	43.91 ± 0.01^cd^	39.37 ± 0.01^de^
PUFA		11.70 ± 0.30^k^	33.10 ± 0.10^h^	3.93 ± 0.04^l^	54.30 ± 0.10^b^	68.00 ± 1.00^a^	nd	36.11 ± 0.02^g^	38.22 ± 0.02^e^

**Table 4 foods-11-03516-t004:** Ergosterol content expressed in mg/100 g dw. For each mushroom sample, means within a column with different letters differ significantly (*p* < 0.05).

Species	Ergosterol
*A. vitellinus*	21.50 ± 0.40^defg^
*C. magellanicus*	38.00 ± 9.00^de^
*C. xiphidipus*	34.00 ± 9.00^def^
*C. aegerita*	72.00 ± 10.00^b^
*C. hariotii*	0.40 ± 0.10^g^
*F. antarctica*	16.00 ± 1.00^efg^
*F. endoxantha*	45.00 ± 1.00^cd^
*F. pumiliae*	71.00 ± 14.00^b^
*F. velutipes*	32.00 ± 2.00^defg^
*G. gargal*	123.57 ± 12.00^a^
*G. sordulenta*	29.00 ± 9.00^def^
*H. dusenii*	16.00 ± 1.00^efg^
*L. nuda*	13.00 ± 0.40^efg^
*L. perlatum*	31.00 ± 1.00^def^
*P. ostreatus*	32.00 ± 2.00^defg^
*R. botrytis*	68.50 ± 0.30^bc^
*R. patagonica*	13.00 ± 1.00^fg^

**Table 5 foods-11-03516-t005:** Phenol content expressed in mg/g. For each mushroom sample, means within a column with different letters differ significantly (*p* < 0.05).

Species	*p*-Coumaric Acid	Gallic Acid	*p*-Hidroxibenzoic Acid	Protocatechuic Acid	3-(3,4-Dihydroxyphenyl)-lactic Acid	Galic Acid Monohidrate	Total Phenols
*A. vitellinus*	nd	1.94 + 0.05^h^	Nd	nd	nd	nd	1.94 + 0.05^i^
*C. magellanicus*	nd	5.79 + 0.23^c^	1.34 + 0.02^e^	0.85 + 0.01^e^	nd	nd	7.90 + 0.20^d^
*C. xiphidipus*	nd	7.36 + 0.07^a^	1.60 + 0.10^d^	1.16 + 0.02^d^	nd	nd	10.20 + 0.10^c^
*C. aegerita*	0.83 + 0.01^b^	4.91 + 0.04^d^	4.10 + 0.10^c^	0.69 + 0.01^f^	nd	1.19 + 0.04	11.70 + 0.10^b^
*C. hariotii*	nd	0.80 + 0.01^j^	Nd	nd	nd	nd	0.80 + 0.01^l^
*F. antarctica*	nd	1.00 + 0.05^j^	Nd	0.15 + 0.01^h^	0.10 + 0.01^c^	nd	1.30 + 0.10^k^
*F. endoxantha*	nd	2.03 + 0.04^h^	5.90 + 0.10^b^	7.65 + 0.04^a^	1.75 + 0.02^a^	nd	11.40 + 0.10^b^
*F. pumiliae*	nd	1.43 + 0.01^i^	0.55 + 0.02^g^	0.69 + 0.01^f^	1.01 + 0.02^b^	nd	3.12 + 0.02^h^
*F. velutipes*	nd	2.84 + 0.04^g^	Nd	nd	nd	nd	2.84 + 0.04^h^
*G. gargal*	nd	3.58 + 0.03^f^	Nd	0.56 + 0.02^g^	nd	nd	4.20 + 0.10^g^
*G. sordulenta*	nd	1.74 + 0.11^hi^	nd	nd	nd	nd	1.70 + 0.10^ij^
*H. dusenii*	nd	4.40 + 0.15^e^	nd	nd	nd	nd	4.40 + 0.12^fg^
*L. nuda*	0.22 + 0.01^c^	3.61 + 0.05^f^	0.87 + 0.02^f^	nd	nd	nd	4.70 + 0.10^f^
*L. perlatum*	4.32 + 0.04^a^	6.75 + 0.13^b^	40.30 + 0.30^a^	nd	nd	nd	51.40 + 0.20^a^
*P. ostreatus*	nd	0.80 + 0.03^j^	0.56 + 0.02^g^		nd	nd	1.40 + 0.10^jk^
*R. botrytis*	nd	3.84 + 0.12^f^	nd	2.90 + 0.10^c^	nd	nd	6.80 + 0.20^e^
*R. patagonica*	nd	2.54 + 0.04^g^	nd	5.30 + 0.07^b^	nd	nd	7.80 + 0.10^d^

**Table 6 foods-11-03516-t006:** Antioxidant activity of the mushroom extracts measured by inhibition of lipid peroxidation (TBARS) and the oxidative hemolysis inhibition assay (OxHLIA). IC_50_ values were expressed in µg/mL. na: no activity (Δ*t* values less than 60 min were obtained). For each mushroom sample, means within a column with different letters differ significantly (*p* < 0.05).

Species	TBARS	OxHLIA
*A. vitellinus*	551.00 ± 9.00^hi^	113.00 ± 7.00^i^
*C. magellanicus*	688.00 ± 268.00^hi^	Na
*C. xiphidipus*	1206.00 ± 200.00^def^	672.00 ± 9.00^b^
*C. aegerita*	2426.00 ± 50.00^ab^	202.00 ± 14.00^g^
*C. hariotii*	1468.00 ± 82.00^cde^	Na
*F. antarctica*	2627.00 ± 189.00^a^	1066.00 ± 81.00^a^
*F. endoxantha*	1132.00 ± 8.00^ef^	335.00 ± 8.00^de^
*F. pumiliae*	1019.00 ± 20.00^fg^	285.00 ± 13.00^ef^
*F. velutipes*	1543.00 ± 100.00^bcd^	426.00 ± 53.00^c^
*G. gargal*	633.00 ± 15.00^h^	376.00 ± 17.00^cd^
*G. sordulenta*	299.00 ± 31.00^ij^	155.00 ± 7.00^h^
*H. dusenii*	610.00 ± 17.00^h^	220.00 ± 11.00^g^
*L. nuda*	711.00 ± 185.00^gh^	93.00 ± 6.00^i^
*L. perlatum*	1217.00 ± 564.00^def^	90.00 ± 4.00^i^
*P. ostreatus*	2052.00 ± 276.00^abc^	Na
*R. botrytis*	167.00 ± 14.00^j^	249.00 ± 12.00^fg^
*R. patagonica*	156.00 ± 12.00^j^	Na
Control/Trolox	18.40 ± 0.10^k^	21.80 ± 0.30^j^

**Table 7 foods-11-03516-t007:** Antimicrobial activity of all the extracts against selected food-contaminating bacteria and fungi. The maximum concentration used was 10 mg/mL. MIC: Minimum inhibitory concentration; MBC: Minimum bactericidal concentration; MFC: Minimum fungicidal concentration; n.t. not tested.

														**Positive Control**					
	**AV**		**CM**		**CX**		**CA**			**CH**				**Streptomicin** **1 mg/mL**		**Methicilin** **1 mg/mL**		**Ampicillin** **10 mg/mL**	
Antibacterial Activity	MIC	MBC	MIC	MBC	MIC	MBC	MIC		MBC	MIC		MBC		MIC	MBC	MIC	MBC	MIC	MBC
**Gram-negative bacteria**																			
*Enterobacter cloacae*	10	>10	10	>10	10	>10	10		>10	10		>10		0.007	0.007	n.t.	n.t.	0.15	0.15
*Escherichia coli*	5	>10	10	>10	10	>10	10		>10	>10		>10		0.01	0.01	n.t.	n.t.	0.15	0.15
*Pseudomonas aeruginosa*	>10	>10	>10	>10	>10	>10	>10		>10	>10		>10		0.06	0.06	n.t.	n.t.	0.63	0.63
*Salmonella enterocolitica*	2.5	>10	5	>10	5	>10	5		>10	10		>10		0.007	0.007	n.t.	n.t.	0.15	0.15
*Yersinia enterocolitica*	1.25	>10	>10	>10	10	>10	>10		>10	10		>10		0.007	0.007	n.t.	n.t.	0.15	0.15
**Gram-positive bacteria**																			
*Bacillus cereus*	10	>10	10	>10	10	>10	>10		>10	10		>10		0.007	0.007	n.t.	n.t.	n.t.	n.t.
*Listeria monocytogenes*	10	>10	10	>10	>10	>10	>10		>10	>10		>10		0.007	0.007	n.t.	n.t.	0.15	0.15
*Staphylococcus aureus*	1.25	>10	5	>10	0.6	>10	1.25		>10	10		>10		0.007	0.007	0.007	0.007	0.15	0.15
														**Ketaconazole** **1mg/mL**					
** Antifungal Activity **	MIC	MFC	MIC	MFC	MIC	MFC	MIC		MFC	MIC		MFC				MIC	MFC		
*Aspergillus brasiliensis*	10	>10	2.5	>10	2.5	>10	2.5		>10	5		>10				0.06	0.125		
*Aspergillus fumigatus*	0.07	0.15	0.07	0.15	0.07	0.15	>10		>10	>10		>10				0.5	1		
														**Positive Control**					
	**FA**		**FE**		**FP**		**FV**			**GG**		**GS**		**Streptomicin** **1 mg/mL**		**Methicilin** **1 mg/mL**		**Ampicillin** **10 mg/mL**	
Antibacterial Activity	MIC	MBC	MIC	MBC	MIC	MBC	MIC		MBC	MIC	MBC	MIC	MBC	MIC	MBC	MIC	MBC	MIC	MBC
**Gram-negative bacteria**																			
*Enterobacter cloacae*	10	>10	>10	>10	10	>10	>10		>10	10	>10	10	>10	0.007	0.007	n.t.	n.t.	0.15	0.15
*Escherichia coli*	10	>10	>10	>10	10	>10	>10		>10	>10	>10	10	>10	0.01	0.01	n.t.	n.t.	0.15	0.15
*Pseudomonas aeruginosa*	>10	>10	>10	>10	>10	>10	>10		>10	>10	>10	>10	>10	0.06	0.06	n.t.	n.t.	0.63	0.63
*Salmonella enterocolitica*	5	>10	10	>10	5	>10	5		>10	10	>10	5	>10	0.007	0.007	n.t.	n.t.	0.15	0.15
*Yersinia enterocolitica*	10	>10	10	>10	5	>10	2.5		>10	2.5	>10	10	>10	0.007	0.007	n.t.	n.t.	0.15	0.15
**Gram-positive bacteria**																			
*Bacillus cereus*	>10	>10	>10	>10	5	>10	5		>10	10	>10	>10	>10	0.007	0.007	n.t.	n.t.	n.t.	n.t.
*Listeria monocytogenes*	10	>10	5	>10	10	>10	10		>10	10	>10	>10	>10	0.007	0.007	n.t.	n.t.	0.15	0.15
*Staphylococcus aureus*	1.25	>10	0.6	>10	0.6	>10	0.6		>10	10	>10	0.3	>10	0.007	0.007	0.007	0.007	0.15	0.15
																**Ketaconazole** **1 mg/mL**			
** Antifungal Activity **	MIC	MFC	MIC	MFC	MIC	MFC	MIC		MFC	MIC	MFC	MIC	MFC			MIC	MFC		
*Aspergillus brasiliensis*	2.5	>10	5	>10	2.5	>10	1.25		>10	5	>10	5	>10			0.06	0.125		
*Aspergillus fumigatus*	>10	>10	0.07	0.15	>10	>10	>10		>10	>10	>10	>10	>10			0.5	1		
														**Positive Control**					
	**HD**		**LN**		**LP**		**PO**		**RB**		**RP**			**Streptomicin** **1 mg/mL**		**Methicilin** **1 mg/mL**		**Ampicillin** **10 mg/mL**	
Antibacterial Activity	MIC	MBC	MIC	MBC	MIC	MBC	MIC	MBC	MIC	MBC	MIC	MBC		MIC	MBC	MIC	MBC	MIC	MBC
**Gram-negative bacteria**																			
*Enterobacter cloacae*	10	>10	>10	>10	10	>10	>10	>10	10	>10	>10	>10		0.007	0.007	n.t.	n.t.	0.15	0.15
*Escherichia coli*	10	>10	10	>10	>10	>10	5	>10	10	>10	10	>10		0.01	0.01	n.t.	n.t.	0.15	0.15
*Pseudomonas aeruginosa*	>10	>10	>10	>10	>10	>10	>10	>10	>10	>10	>10	>10		0.06	0.06	n.t.	n.t.	0.63	0.63
*Salmonella enterocolitica*	5	>10	5	>10	10	>10	5	>10	5	>10	5	>10		0.007	0.007	n.t.	n.t.	0.15	0.15
*Yersinia enterocolitica*	5	>10	5	>10	>10	>10	2.5	>10	1.25	>10	5	>10		0.007	0.007	n.t.	n.t.	0.15	0.15
**Gram-positive bacteria**																			
*Bacillus cereus*	10	>10	5	>10	5	>10	>10	>10	10	>10	5	>10		0.007	0.007	n.t.	n.t.	n.t.	n.t.
*Listeria monocytogenes*	>10	>10	10	>10	5	>10	10	>10	>10	>10	10	>10		0.007	0.007	n.t.	n.t.	0.15	0.15
*Staphylococcus aureus*	0.6	>10	0.6	>10	2.5	>10	0.6	>10	0.3	>10	2.5	>10		0.007	0.007	0.007	0.007	0.15	0.15
																**Ketaconazole** **1 mg/mL**			
** Antifungal Activity **	MIC	MFC	MIC	MFC	MIC	MFC	MIC	MFC	MIC	MFC	MIC	MFC				MIC	MFC		
*Aspergillus brasiliensis*	1.25	>10	2.5	>10	2.5	>10	2.5	>10	5	>10	5	>10				0.06	0.125		
*Aspergillus fumigatus*	0.07	0.15	>10	>10	>10	>10	>10	>10	>10	>10	0.07	0.15				0.50	1		

## Data Availability

The data presented in this study are available upon request from the corresponding author.

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
