# Peer review of "Nutritional Composition and Bioactive Properties of Wild Edible Mushrooms from Native *Nothofagus* Patagonian Forests"

_foods, 2022, doi:10.3390/foods11213516_

Round 1
Reviewer 1 Report
1.Change the key words ,2. Expand the words coming first time in text , then follow abbreviation .,3. How the authors are sure that , the explored specieses are not poisonous? 4. Fall and spring season mushrooms has different nutritional value, then why not given separate identity to them?
5. Why the samples were not preserved a-in deep freezer? samples were stored at -18 degree C . It may affect the actual values of nutrients.
6. The manuscript seems too lengthy. The Antimicrobial part may be taken as separate paper.

Author Response
Change the key words
Answer: Thank you for your careful reading of the manuscript and for all your comments. All suggestions for changes were made to guarantee the improvement of the article. New keywords have been added to replace some of the ones initially selected.
- Expand the words coming first time in text, then follow abbreviation.
Answer: We appreciate the comment. The manuscript has been carefully revised to fully identify the meaning of the first times the abbreviations appear.
- How the authors are sure that, the explored specieses are not poisonous?
Answer: Most of these species were frequently mentioned by Mapuche communities as having edible value. The selected list of understudied species have consumption records, reported in previous works. We do not include any species with doubtful or scant information.
- Fall and spring season mushrooms has different nutritional value, then why not given separate identity to them?
Answer: Fruiting is associated with the autumn rains, given the marked seasonality of Patagonia. The fungi used in this study grow exclusively in the fall (except Cyttaria). Others wild edible mushrooms grow in spring (principally Morchella genus), but we do not included in this study. No comparison could be made attending seasons, as species change with seasons.
- Why the samples were not preserved a-in deep freezer? samples were stored at -18 degree C. It may affect the actual values of nutrients.
Answer: We used previous reports and methodology in which samples were dried stored until the extract preparation. Further studies should analyze how nutrient values change with different preservation methods.
- The manuscript seems too lengthy. The Antimicrobial part may be taken as separate paper
Answer: We consider important to maintain the results of the antimicrobial analyzes in this work. It supports the medicinal potentiality of WEM described.
Reviewer 2 Report
1. Revise the headings
2. Add the application of study in abstract
3. Revise keyword list and avoid those in the manuscript title
4. Enlarge the manuscript introduction
5. Add related sample pictures
6. Add most relevant and latest references
7. Keep results and discussion together
8. Add mushrooms pictures
9. Check for grammatical errors
10 Check for technical and typo error
Author Response
- Revise the headings
Answer: Thank you for your careful reading of the manuscript and for all your comments. All suggestions for changes were made to guarantee the improvement of the article.
- Add the application of study in abstract
Answer: The application of the study is presented in the abstract “Having nutritional and nutraceutical information about WEM will help to incorporate them into a functional diet, which chooses them as safe, nutritious and healthy foods, and to use them in an identity mycogastronomy linked to tourism development.”
- Revise keyword list and avoid those in the manuscript title
Answer: As requested, keywords have been revised and changed.
- Enlarge the manuscript introduction
Answer: We consider the introduction already addresses the issues necessary to present the study. As previous studies regarding chemical and nutritional profiles of these wild mushroom species are scant, we do not find other important topics to include.
- Add related sample pictures
Answer: We appreciate your suggestion and added Figure 1 to the manuscript that contains photographs of all species analyzed in the present study.
- Add most relevant and latest references
Answer: More than 50% of references are of new works (last 10 years) related with wild edible mushrooms and nutritional aspects.
- Keep results and discussion together
Answer: In this manuscript, the results and the discussion were presented in the same section
- Add mushrooms pictures
Answer: The photographs referring to the analyzed species were compiled and presented in Figure 1 added to the manuscript
- Check for grammatical errors
Answer: Detected grammatical error were corrected.
10 Check for technical and typo error
Answer: Detected error were corrected.
Reviewer 3 Report
I revised the manuscript about the chemical an some bioactive properties of 17 edible mushroom in Nothofagus Patagonian forests. In the study, a heavy work load is exist. The detailed characterizations were done for those edible mushrooms but the text is lack of discussions. The results were directly given and compared with the results previously reported in the literature. Any discussions were not done in the text regarding the reasons of the wide variations in each determined parameters in different mushroom species. Moreover, correlation analysis, PCA or other statistical analysis should have done to reveal the relations between the determined parameters.
In abstract: What is WEM?
Line 36. Please seperate the words of “aminoacids”.
Please discuss the possible reasons of wide variations in total phenolic, ash, and carbohydrate content between the edible mushroom.
The unit of organic acid results should be checked. g/100g representation seems incorrect.
In Table 2, how did you determine the amount of non-identified sugar without using a calibration curve of standard molecule or compound?
Tbale 3 says “Fatty acids composition (%) of the studied wild mushrooms on a dry weight basis”, but the results seems that given in different expression unit.
In lne 429. Gallic acid was detected in all studied mushrooms, in a range between 0.80 and 7.36 g/g dw, in C. hariotii and C. xiphidipus respectively. But ın Table 5, ın the headline it says “Phenol content expressed in mg/g”. Please correct the unşts in the text.
Table 7 can not be seen on the text.
Author Response
I revised the manuscript about the chemical an some bioactive properties of 17 edible mushroom in Nothofagus Patagonian forests. In the study, a heavy work load is exist. The detailed characterizations were done for those edible mushrooms but the text is lack of discussions. The results were directly given and compared with the results previously reported in the literature. Any discussions were not done in the text regarding the reasons of the wide variations in each determined parameters in different mushroom species. Moreover, correlation analysis, PCA or other statistical analysis should have done to reveal the relations between the determined parameters.
Answer: Thank you for your careful reading of the manuscript and for all your comments. All suggestions for changes were made to guarantee the improvement of the article.
In abstract: What is WEM?
Answer: The abbreviation means “Wild edible mushrooms”. The full mention was added to the text the first time the abbreviation appears
Line 36. Please separate the words of “aminoacids”.
Answer: The suggested change was made.
Please discuss the possible reasons of wide variations in total phenolic, ash, and carbohydrate content between the edible mushroom.
Answer: The relevant information that justifies these data was added in the manuscript “Variation in nutrients and bioactive properties is common between mushroom species. Chemical composition of mushroom species may be affected by several variables such as genetic structure, strains, maturation stage, environmental conditions, such as soil composition, as well as the specific part of the mushroom, postharvest preservation method (dry or fresh procedures), and cooking process (Barroetaveña & Toledo, 2017; Kalac, 2013).”
The unit of organic acid results should be checked. g/100g representation seems incorrect.
Answer: The manuscript has been carefully revised and the units have been corrected.
In Table 2, how did you determine the amount of non-identified sugar without using a calibration curve of standard molecule or compound?
Answer: There are several studies in the literature that report the presence of this sugar molecule in this type of matrices, however, our research team does not have the standards so that its identification can be confirmed with certainty. In this way, quantification was performed based on the standard referring to the closest molecule in terms of retention time.
Table 3 says “Fatty acids composition (%) of the studied wild mushrooms on a dry weight basis”, but the results seems that given in different expression unit.
Answer: The manuscript has been carefully revised and the results of fatty acids were expressed as a relative percentage (%).
In line 429. Gallic acid was detected in all studied mushrooms, in a range between 0.80 and 7.36 g/g dw, in C. hariotii and C. xiphidipus respectively. But ın Table 5, ın the headline it says “Phenol content expressed in mg/g”. Please correct the unşts in the text.
Answer: The manuscript was checked, and the requested change was made.
Table 7 can not be seen on the text.
Answer: The reference to table 7 is included in line 485.
Round 2
Reviewer 1 Report
Author made sincere effort in improving the manuscript. Seems in order.
1. The author aptly replied on
a) How the seasonal variation in the nutritional profile of the mushroom was taken care of ?
b) Safety issues- How they ascertained that the mushrooms included in the study were non-poisonous?
2. The study relevant and interesting
3. High degree of originality is involved in this study
4. The study add to the subject area compared with other published material, the studied strains are different than the previous ones.
5. May reduce the content in introduction and conclusion part by eliminating non -essential description.
6. By and large it is good , but author may reconstruct the very lengthy sentences.
7. The conclusions Too lengthy, needs to concise.
8. The study address the main question posed.
Author Response
Reviewer #1: 1. Author made sincere effort in improving the manuscript. Seems in order.
Answer: We appreciate your review and your comments that allowed the improvement of the manuscript.
- The author aptly replied on
- a) How the seasonal variation in the nutritional profile of the mushroom was taken care of?
Answer: each species was collected in its fruiting season; all of them only fruit in autumn or spring. We add a reference that describes fruiting phenology by species in the manuscript, and explained better the fruiting seasons. Within the fruiting season, our sample did not account for moment of collection. That could be a question for further studies; anyway, fruiting seasons are short and pretty stable in the Patagonian Andes of Argentina, because of the Mediterranean situation: fruiting starts with strictly seasonal rains and end with freezing temperatures (in winter) or drought (in late spring).
- b) Safety issues- How they ascertained that the mushrooms included in the study were non-poisonous?
Answer: as it was detailed in the text, all the studied species have register of effective consumption. To determine the edibility of a particular species, several works in the field of
ethnomycology, which address aspects related to cultural perception, classification
and traditional use of fungi, show that effective consumption is the given evidence
of edibility (e. g. Ruan-Soto et al. 2007; Garibay-Orijel et al. 2006, 2007) and possibly potential for consumption, if the registry is very isolated.
Ruan-Soto F, Mariaca R, Cifuentes J, Limón F, Pérez-Ramírez L, Sierra-Galván S (2007) Nomenclatura, clasificación y percepciones locales acerca de los hongos en dos comunidades de la Selva Lacandona, Chiapas, México. Etnobiología 5:1–20
Garibay-Orijel R, Cifuentes J, Estrada-Torres A, Caballero J (2006) People using macro-fungal diversity in Oaxaca. Mexico Fungal Divers 21:41–67
Garibay-Orijel R, Caballero J, Estrada-Torres A, Cifuentes J (2007) Understanding cultural significance, the edible mushrooms case. J Ethnobiol Ethnomed 3:1–18
- The study relevant and interesting
Answer: We appreciate your comment.
- High degree of originality is involved in this study
Answer: We appreciate your comment.
- The study add to the subject area compared with other published material, the studied strains are different than the previous ones.
Answer: We appreciate your comment.
- May reduce the content in introduction and conclusion part by eliminating non -essential description.
Answer: We decided to add text (short sentences with more precise information) in the introduction as other reviewer suggested it was very brief. We consider your comment and shortened the conclusion.
- By and large it is good, but author may reconstruct the very lengthy sentences.
Answer: we shortened long sentences
- The conclusions Too lengthy, needs to concise.
Answer: we shortened the conclusion
- The study address the main question posed.
Answer: We appreciate your comment.

Reviewer 3 Report
Dear authors,
I am so pleased that you revised and improved the manuscript which has the potentail to make scientific contribution to the research area.
Author Response
Reviewer #2: 1. Dear authors, I am so pleased that you revised and improved the manuscript which has the potentail to make scientific contribution to the research area.
Answer: Thank you for your careful reading of the manuscript and for all your comments that allowed the manuscript to be improved.